# Immunophenotypical Characterization of M1/M2 Macrophages and Lymphocytes in Cisplatin-Induced Rat Progressive Renal Fibrosis

**DOI:** 10.3390/cells10020257

**Published:** 2021-01-28

**Authors:** Minto Nakagawa, Mohammad Rabiul Karim, Takeshi Izawa, Mitsuru Kuwamura, Jyoji Yamate

**Affiliations:** 1Laboratory of Veterinary Pathology, Graduate School of Life and Environmental Sciences, Osaka Prefecture University, 1-58 Rinku-Ourai-Kita, Izumisano City, Osaka 598-8531, Japan; minto_0207@yahoo.ne.jp (M.N.); mrabiulkarim@bau.edu.bd (M.R.K.); izawa@vet.osakafu-u.ac.jp (T.I.); kuwamura@vet.osakafu-u.ac.jp (M.K.); 2Department of Anatomy and Histology, Faculty of Veterinary Science, Bangladesh Agricultural University, Mymensingh 2202, Bangladesh

**Keywords:** cisplatin, M1/M2 macrophages, T lymphocytes, renal fibrosis, rats

## Abstract

Renal fibrosis is regarded as the common final pathway leading to chronic kidney diseases; macrophages and myofibroblasts play important roles in the development of fibrosis. F344 rats were injected once with cisplatin (CDDP; 6 mg/kg BW) for renal lesions. Here, immunophenotypical characteristics of macrophages and lymphocytes in CDDP-induced rat renal lesions were investigated histopathologically; the CDDP-induced renal lesions consisted of tissue damage at the early-stage, worsen the damage and commencement of interstitial fibrosis at the mid-stage, and progressive fibrosis at the late stage; the KIM-1 expression and α-SMA^+^ myofibroblast area reflected renal tubular damage/abnormal regeneration and renal interstitial fibrosis, respectively. CD68^+^ M1 macrophages began to increase at the mid-stage, with increased mRNA expressions of M1-related cytokines (INF-γ, TNF-α and IL-6), and then slightly decreased at the late-stage. CD163^+^ M2 macrophages showed a gradually increased number at the mid- and late-stages, accompanied by increased TGF-β1 mRNA expression (a fibrogenic factor). Double immunofluorescence using fibrotic samples at the late-stage revealed that 62.0–78.0% of CD68^+^ M1 macrophages co-expressed CD163, indicating that M1/M2 macrophages may contribute to progressive renal fibrosis in cooperation; further, MHC class II-expressing macrophages had a tendency towards M1 polarization, whereas CD204-expressing macrophages towards M2 polarization. In addition, CD4^+^ and CD8^+^ T cells were increased at the late-stage. Collectively, progressive renal interstitial fibrosis may be developed by complicated mechanisms that arose via interaction of M1/M2 macrophages (inflammatory for M1 and anti-inflammatory for M2) and T cells reacting to CD4 (for helper) and CD8 (for cytotoxicity). This study would provide some information on the pathogenesis of renal fibrosis based on inflammatory cells.

## 1. Introduction

Regardless of initial causes, renal fibrosis is regarded as the common final pathway leading to chronic kidney diseases (CKD). The lesion is characterized histopathologically by an excessive accumulation and deposition of extracellular matrices (ECMs) in the interstitium. CKD ultimately results in renal dysfunction and end-stage renal failure, and the patients require dialysis or transplantation. The pathogenic mechanism of renal fibrosis should be clarified to prevent the progression.

Generally, macrophages and myofibroblasts play important roles in the development of fibrosis [1,2]. Myofibroblasts are mesenchymal cells capable of producing ECMs, and transforming growth factor-β1 (TGF-β1) is an important factor for the activation of myofibroblasts [3]. Macrophages produce TGF-β1 [4]. In renal fibrosis, TGF-β1 may promote tubular epithelial cell transdifferentiation into myofibroblasts (epithelial-mesenchymal transition; EMT) and enhance interstitial deposition of ECMs produced by myofibroblasts [5,6]. Therefore, macrophages are crucial in the pathogenesis of renal fibrosis [1]. Moreover, macrophages can contribute to the promotion or resolution of inflammation, removal of apoptotic cells by phagocytosis, and support of cell proliferation following tissue injury, indicating that macrophages have heterogeneous functions [7,8].

Like Th1 and Th2 in T helper cells, macrophages are classified as M1 (classically activated) and M2 (alternatively activated) types [9]. M1 macrophages are induced by interferon-γ (IFN-γ) and are characterized by active production of inflammatory cytokines, reactive nitrogen and oxygen intermediates, and promotion of Th1 response. In contrast, M2 macrophages are activated by IL-4 and associated with the resolution of inflammation and promotion of tissue remodeling and fibrosis [10]. Such macrophage plasticity is called M1/M2 polarization. Generally, the change of M1/M2 phenotypes of macrophages easily occurs in vitro; however, such events may not always be corresponding to that in pathological lesions in vivo because there are complicated factors in the body that should influence the functions of macrophages [11]. The histopathological/immunohistochemical approach of M1/M2-macrophages in vivo is very important.

To investigate the histopathological alterations of macrophages in pathological settings, therefore, the analysis of expression profiles of cell surface molecules are useful [9]. In the present study, by using renal fibrosis lesions induced in rats by cisplatin with renal toxicity, we performed the immunohistological phenotyping of macrophages using antibodies against cell surface molecules. CD68 is an antigen on lysosomal and endosomal membranes related to phagocytic activities [12], and its expression is regarded as an M1 macrophage marker [13,14]. CD163 is a receptor for hemoglobin-haptoglobin complexes and a member of the scavenger receptor cysteine-rich group B family. CD163 expression is regarded as a marker for M2 macrophages [15].

Although the appearance of macrophages was observed in rat renal fibrosis [16,17], the immunohistochemical characteristic of macrophages and lymphocytes have not yet been analyzed on the basis of the concept of M1/M2-macrophage polarization. The concept of M1/M2 polarization may be a key for understanding the pathogenesis of renal fibrosis, which would provide a basis for macrophage-centered diagnostic and therapeutic strategies [10]. We also investigated immunohistochemically the participation of lymphocytes, which may be related to M1/M2 polarization. Our results suggest that M1/M2 macrophages, as well as T cells, may contribute to the progressive renal fibrogenesis after tubular injury.

## 2. Materials and Methods

### 2.1. Experimental Procedures

Five-week-old male F344/DuCrj rats (106–146 g; Charles River Japan, Hino, Shiga, Japan) were housed in an animal room controlled at 22 ± 3 °C and with a 12 h light–dark cycle and fed a standard diet (DC-8, CLEA Japan, Inc., Tokyo, Japan) and filtered tap water ad libitum. After one-week acclimatization, twenty-four rats received a single intraperitoneal injection of cisplatin (CDDP, cis-diamminedichloroplatinum; Nippon Kayaku Co. Ltd., Tokyo, Japan) at a dose of 6 mg/kg body weight. Three rats were examined on each of days 1, 3, 5, 7, 9, 12, 15 and 20 after CDDP injection. Three rats as control received an equal volume of physiological saline in the same manner and were examined on day 0. All rats were euthanized under deep isoflurane anesthesia. Experimental protocols were performed according to the Institutional Guidelines for Animal Care and Use (Nos. 23–24 and 25–85) in Osaka Prefecture University.

### 2.2. Histopathology and Immunohistochemistry

Renal tissues were immediately fixed in 10% neutral buffered formalin or periodate-lysine-paraformaldehyde (PLP) solution and embedded in paraffin with AMeX (acetone, methyl benzoate and xylene) method [18]. Deparaffinized formalin-fixed sections were stained with hematoxylin and eosin (HE) for morphological examination and with the azan–Mallory stain for collagen deposition. Deparaffinized PLP-fixed sections were immune-stained with primary antibodies and peroxidase-conjugated secondary antibody (Histofine simple stain MAX-PO; Nichirei, Tokyo, Japan). Details of the primary antibodies are listed in Table 1. Tissue sections were immunostained by Histostainer (Nichirei). Positive reactions were visualized with 3,3′-diaminobenzidine tetrahydrochloride (DAB; Nichirei), and the sections were then lightly counterstained with hematoxylin. For negative controls, tissue sections were treated with mouse or rabbit non-immunized serum instead of the primary antibody. Positive cells were counted in five randomly selected 40 × fields in the corticomedullary junction in which interstitial fibrosis occurs in this rat model [19]. Areas of myofibroblasts reacting to α-SMA (the most reliable marker for this cell type) were measured in five different fields of the corticomedullary junction using an image analyzing software (WinROOF, Mitani Corp., Fukui, Japan); because blood vessel-constituting smooth muscle react to α-SMA [20], blood vessels were excluded for the analysis. In addition, a semiquantitative evaluation was made for interstitial fibrosis degree based on HE-stained and azan-Mallory-stained sections: no-fibrosis (−), slight degree (1+), moderate degree (2+), severe degree (3+) and more severe degree (4+) in the affected corticomedullary region [20].

### 2.3. Double Immunofluorescence

Ten-μm fresh-frozen sections from the kidney of control and CDDP-injected rats were used for double immunofluorescence; the antibody pairs were CD68 in combination with CD163, MHC class II and CD204, as well as CD163 with MHC class II and CD204. After fixation in cold PLP solution for CD68/CD204 combination and in cold acetone: methanol (1:1) solution for other combinations, the sections were incubated with 10% normal goat serum for 30 min, followed by reaction with the primary antibody overnight at 4 °C. After rinsing in PBS, the sections were incubated for 45 min with the goat anti-mouse IgG-conjugated with Alexa 568 (Invitrogen, Carlsbad, CA, USA). Then, the sections were incubated at room temperature for an hour with the following second primary antibodies labeled with fluorescent dye: Alexa 488-labeled CD68 (AbD Serotec, Oxford, UK) for CD68/CD163 and CD68/CD204 combinations; Alexa 488-labeled CD163 (AbD Serotec, Oxford, UK) for CD163/CD204; Alexa 488-labeled MHC class II (AbD Serotec, Oxford, UK) forCD68/MCH class II and CD163/MHC class II. The sections were cover-slipped with Vectashield^TM^ mounting medium containing 4′,6-diamidino-2-phenylindole (DAPI) (Vector Laboratories Inc., Burlingame, CA, USA) for nuclear staining and analyzed using a virtual slide scanner (VS-120; Olympus, Tokyo, Japan). Cells co-expressing CD68/CD163, CD68/MHC class II, CD163/MHC class II, CD68/CD204, and CD163/CD204 were counted in five randomly selected 40× fields of the corticomedullary junction.

### 2.4. Reverse-Transcription Polymerase Chain Reaction (RT–PCR)

Renal tissues collected from the corticomedullary junction were immediately immersed in RNAlater reagent (Qiagen, Hilden, Germany), kept overnight at 4 °C and stored at −80 °C until use. Total RNA was extracted from renal tissues using an SV Total RNA isolation system (Promega, Madison, WI, USA) according to the manufacturer’s instructions. Two and a half µg of total RNA was reverse-transcribed with Superscript VILO reverse transcriptase (Life Technologies, CA, USA). Real-time PCR was performed using Thunderbird Probe qPCR mix (Toyobo Co. Ltd., Osaka, Japan) with TaqMan gene expression assays (Life Technologies, Carlsbad, CA, USA) or Thunderbird SYBR qPCR mix (Toyobo) in a PikoReal real-time 96 PCR system (Thermo Scientific, Waltham, MA, USA). Taqman probes for IL-6 (Assay ID: Rn01410330_m1), IL-10 (Assay ID: Rn00563409_m1), IL-4 (Assay ID: Rn01456866_m1), andβ-actin (Assay ID: Rn00667869_m1) were used. The primers used in the SYBR Green real-time PCR system are listed in Table 2. The data were calculated using the comparative C_t_ method (^ΔΔ^C_t_ method) and normalized by the expression of β-actin mRNA as an internal control.

### 2.5. Statistical Analysis

Data were statistically analyzed by the one-way analysis of variance (ANOVA) followed by Dunnett′s multiple comparison test to investigate the kinetics against controls. Values of *p* < 0.05 were considered significant from controls [14].

## 3. Results

### 3.1. Histopathology of CDDP-Induced Rat Renal Lesion

Control kidneys showed normal renal histological architecture (Figure 1A). On days 1 and 3 after CDDP injection, renal proximal tubular epithelial cells, especially at the S3 segment in the corticomedullary junction, underwent swelling with nuclear degeneration (Figure 1B); the degeneration was more clearly seen on day 3. On days 5 and 7, renal epithelial cells of the affected tubules were desquamated (Figure 1C,D), and then, the tubules were variously dilated, being lined by regenerating flattened or cuboidal epithelial cells. In addition, inflammatory cells began to be seen around the affected renal tubules and their lumina. On days 9 (Figure 1E), 12, and 15, the regenerating epithelial cells showed variable morphology (flattened, polyhedral, and cuboidal), indicating abnormal regeneration; around the variously dilated renal tubules, interstitial fibrosis was developed, accompanied by inflammatory cells. On day 20, interstitial fibrosis was more progressive, particularly around the renal tubules with incompletely regenerating epithelial cells (Figure 1F); inflammatory cells were continuously observed.

mRNA expression of kidney injury molecule-1 (KIM-1), a well-known biomarker for renal proximal tubule injury [21,22], was increased significantly on days 5–15 (Figure 2A), being consistent with histological tubular damage. The immunoreaction to KIM-1 was seen mainly in cuboidal or flattened epithelial cells in the damaged renal tubules (Figure 2B). The KIM-1 expression reflected the histopathological findings of renal tubular damages/abnormal regeneration.

### 3.2. Collagen Deposition and Myofibroblast Appearance in CDDP-Induced Rat Renal Lesion

In the kidneys of control and on days 1 and 3, interstitial fibrosis was not seen (fibrosis degree, −) in the affected corticomedullary junction. Collagen fibers stained blue with the azan–Mallory method began to be gradually seen around the affected renal tubules on days 5 (−~1+), 7 (1+~2+), 9 (2+~3+), and 12 (2+~3+); the deposition degree became gradually severe on days 15 (3+~4+) and 20 (3+~4+) (Figure 3A,B). Immunohistochemically, on day 7, α-SMA^+^ myofibroblasts began to clearly appear in the interstitium around the affected renal tubules lined by abnormal regenerating epithelial cells. On day 9 onwards, the area of α-SMA^+^ myofibroblasts showed a significant increase with a peak on day 15 (Figure 3C,D), in comparison to controls. The area of α-SMA-positive myofibroblasts sufficiently reflected the degrees of progressive interstitial fibrosis.

Although the lesions were overlapping to each other, based on the renal tubular damage/interstitial fibrosis, the renal failure induced by CDDP injection were regarded as following: tissue damage/cell injury at the early-stage on days 1–3, abnormal regeneration of damaged tubules and commencement of fibrosis at the mid-stage on days 5–9, and progressive fibrosis at the late-stage on days 12–20.

### 3.3. Macrophages Appearing in CDDP-Induced Rat Renal Lesion

Along with CD68^+^ (M1) and CD163^+^ (M2) macrophages, macrophages reacting to MHC class II and CD204 were evaluated in the CDDP-induced renal lesion.

#### 3.3.1. CD68^+^ Macrophages

In the control and affected kidneys on days 1 and 3, a small number of CD68^+^ macrophages were seen in the corticomedullary junction (Figure 4A and Figure 5A). The number of CD68^+^ macrophages increased significantly from day 5 onwards, with a peak on day 9 (Figure 4A). CD68^+^ macrophages were distributed around the damaged renal tubules and within their lumen (Figure 5B,C).

#### 3.3.2. CD163^+^ Macrophages

A few CD163^+^ macrophages were observed in the control and affected kidneys on days 1 and 3 (Figure 4B and Figure 5D). The number of CD163^+^ macrophages was significantly increased on days 5 to 20; that is, a large number of CD163^+^ cells were distributed mainly in the interstitium of the renal medulla (Figure 5E,F); the positive cells were rarely seen within the damaged renal lumen.

#### 3.3.3. MHC Class II^+^ Macrophages

A few MHC class II^+^ macrophages were seen in the control and affected kidneys on days 1 to 7 (Figure 4C and Figure 5G). On day 9, the number of MHC class II^+^ macrophages was quickly increased in the corticomedullary junction, showing the statistical significance (Figure 4C and Figure 5H). On days 12 to 20, the number of MHC class II^+^ macrophages retained increased, and the positive cells were seen exclusively around the damaged renal tubules (Figure 5I).

#### 3.3.4. CD204^+^ Macrophages

In the control and affected kidneys on days 1 and 3, only a few CD204^+^ macrophages were seen (Figure 4D and Figure 5J). The number of CD204^+^ macrophages was significantly increased on days 5 to 12 (Figure 4D and Figure 5K). The number was then decreased to the control level on days 15 and 20 (Figure 4D and Figure 5L). CD204^+^ macrophages were seen mainly around the affected renal tubules and within their lumen (Figure 5K).

### 3.4. M1/M2 Macrophage Polarization in CDDP-Induced Rat Renal Lesion

In order to investigate the macrophage polarization for M1 (CD68) or M2 (CD163) type, double immunofluorescence with combinations of CD163/CD68 (Figure 6A), CD68/MHC class II (Figure 6B), CD163/MHC class II (Figure 6C), CD68/CD204 (Figure 6D), and CD163/CD204 (Figure 6E) were conducted in the affected kidneys at the late-stage, because progressive fibrosis with increasing macrophage number was seen mainly on days 12, 15 and 20. On the examination points, the percentage of CD163^+^/CD68^+^ cells to CD68^+^ cells was from 62.0 to 78.0%; CD68^+^/MHC class II^+^ cells to MHC class II^+^ cells ranged in percentage from 33.2 to 80.3%, whereas CD163^+^/MHC class II^+^ cells to MHC class II^+^ cells from 9.8 to 16.4%, indicating the polarization of MHC class II^+^ macrophages toward CD68^+^ M1 type; CD68^+^/CD204^+^ cells to CD204^+^ cells ranged in percentage from 25.0 to 45.5%, whereas CD163^+^/CD204^+^ cells CD204^+^ cells from 71.2 to 80.0%, indicating that CD204^+^ macrophage had a tendency to CD163^+^ M2 polarization.

### 3.5. M1- and M2-Related Factors in CDDP-Induced Rat Renal Lesion

Renal expression of IFN-γ, TNF-α, and IL-6 as M1-related inflammatory factors and TGF-β, IL-10, and IL-4 as M2-related anti-inflammatory factors [1,23] was analyzed.

mRNAs of IFN-γ (Figure 7A) and TNF-α (Figure 7B) significantly increased on days 5 to 15 and on days 5 and 7, respectively. IL-6 mRNA (Figure 7C) increased earlier than that of IFN-γ and TNF-α; the expression significantly increased on days 3, 5 and 9. TGF-β1 mRNA significantly increased on days 7 to 20 (Figure 7D). mRNAs of IL-10 and IL-4 did not show any significant changes.

### 3.6. Lymphocyte Appearance in CDDP-Induced Rat Renal Lesion

#### 3.6.1. CD4^+^ T Lymphocytes

In the control and affected kidneys on day 1, a small number of CD4^+^ T cells were sporadically seen without showing significant changes (Figure 8A and Figure 9A). The number of CD4^+^ T cells was significantly increased on days 3 to 20; in particular, the markedly increased number retained on day 9 onwards (Figure 8A), and the positive cells appeared mainly around the damaged renal tubules (Figure 9B,C).

#### 3.6.2. CD8^+^ T Lymphocytes

A few CD8^+^ T cells were seen in the control and affected kidneys on days 1–7, without significant changes (Figure 8B and Figure 9D). Then, the number of CD8^+^ T cells showed a significant increase on days 9 to 20 (Figure 8B), and the positive cells were observed diffusely or focally around the damaged renal tubules (Figure 9E,F).

#### 3.6.3. CD20^+^ B Lymphocytes

There were a small number of CD20^+^ B cells in the control and affected kidneys on days 1 and 3 (Figure 8C and Figure 9G). On days 5 and 7, the number of CD20^+^ B cells was significantly increased, showing a transient increase (Figure 8C and Figure 9H); however, the positive cell number was much less than that of CD4^+^ and CD8^+^ T cells. Thereafter, CD20^+^ B cells were suddenly decreased to control levels on day 9 onwards (Figure 8C and Figure 9I).

## 4. Discussion

Experimentally induced rat renal fibrosis by CDDP injection has been used, mainly focusing on injured renal tubules and interstitial fibrosis [16,19,20,24]. The degrees and progression of interstitial fibrosis seen in the present rat renal failure were generally corresponding to those reported previously [16,20]. Renal fibrosis is characterized by the appearance of myofibroblasts [20]. The myofibroblasts reacting to α-SMA was increased with advancing renal fibrosis in CDDP-injected rats [20], as seen in the present study. It is reported that there are myofibroblasts co-expressing to α-SMA/vimentin or α-SMA/desmin, indicating that myofibroblasts show various cytoskeletons; in addition, renal myofibroblasts express Thy-1 (CD90; immature stromal stem marker) [20]. Because the pericytes show a positive reaction to Thy-1, renal myofibroblasts may be originated partly from immature mesenchymal cells [20]. Further, the myofibroblasts may be developed via the EMT evoked from incompletely regenerating renal tubules [24]. The CDDP-induced rat renal fibrosis resembles those of human renal progressive fibrosis like CKD [20].

The pathogenesis of renal fibrosis remains to be fully investigated [25], particularly in terms of inflammatory cell reaction. In human CKD and renal failure animal models, macrophage infiltration is a common pathological feature [26]. Macrophages may play an important role in the pathogenesis. The present immunohistochemical results in CDDP-induced rat renal lesions showed that M1 macrophages reacting to CD68 began to increase on day 5, with a peak on day 9; thereafter, the CD68^+^ M1 macrophages gradually decreased until day 20. Expression of M1-related inflammatory factors such as IFN-γ, TNF-α and IL-6 began to increase on days 3 and 5 and retained increased mainly at the mid-stage. The appearance of CD68^+^ M1 macrophages generally corresponded to increased M1-related inflammatory factors. Histopathologically, the injury of renal epithelial cells by CDDP began to be seen clearly on day 3, and the injury lesions became greater in severity on days 5 and 7; the damaged renal tubules were characterized by swelling, necrosis and desquamation of epithelial cells [19,20]. Additionally, mRNA expression of KIM-1, a marker of renal injury [27], began to significantly increase on day 5, with the continuous expression until day 15; KIM-1 immuno-expression was seen mainly in abnormally regenerating epithelial cells of damaged tubules. Collectively, CD68^+^ macrophages appearing mainly at the mid-stage may participate in the progressive renal injury which was caused initially by CDDP. It is known that M1 macrophages promote tissue injury by producing inflammatory factors [9,28].

Generally, fibrosis after tissue injury is regarded as a healing process [25]. In the present renal failure by CDDP, renal interstitial fibrosis began to be seen clearly on days 7 (fibrosis degree 1+~2+) and 9 (fibrosis degree 2+~3+) and subsequently occurred progressively on days 12 (fibrosis degree 2+~3+), 15 (fibrosis degree 3+~4+) and day 20 (fibrosis degree 3+~4+). The interstitial fibrosis degrees corresponded to the appearance of α-SMA^+^ myofibroblasts. Interestingly, CD163^+^ M2 macrophages began to increase on day 5 and retained the increased number until day 20. It is considered that in contrast to M1 macrophages, M2 macrophages have roles of anti-inflammatory functions and fibrogenesis [9,25]. TGF-β1, which is the most powerful factor for fibrogenesis [3,25], was increased on days 7 to 20, being generally similar to the appearance of CD163^+^ M2 macrophages. Collectively, progressive renal interstitial fibrosis in the present study at the mid and late-stages on days 9 to 20 may be associated with the appearance of CD163^+^ M2 macrophages and increased TGF-β1.

It has been considered that M1/M2 macrophage phenotypes could be interchangeable depending on microenvironments [7,10,14]. Double immunofluorescence with CD68/CD163 showed that CD68^+^ macrophages, which simultaneously expressed CD163, were seen at the late-stage with the development of fibrosis. It is likely that M1 macrophages could undergo a phenotypic switch towards M2 macrophages. The appearance of CD68^+^/CD163^+^ macrophages with both phenotypes of M1/M2 types may have attributed to progressive renal damage by showing biphasic functions of tissue injury (for M1 type) and fibrosis (for M2 type). Previously, it was reported that M1-/M2-macrophages contribute in a cooperated manner to the development of pseudo-lobules (surrounded by fibrotic lesions) in chemically induced rat cirrhosis [13]. In order to investigate the M1/M2 polarization of MHC class II^+^ and CD204^+^ macrophages, the double immunofluorescence with combinations of CD68/MHC class II and CD163/MHC class II or CD68/CD204 and CD163/CD204 were conducted. Particularly, at the end stages on days 9 to 20, the number of MHC class II^+^ macrophages expressing concomitantly CD68 was much greater than that of CD163^+^/MHC class II^+^ macrophages. Apparently, MHC class II^+^ macrophages showed a tendency of M1 polarization. On the contrary, the number of CD68^+^/CD204^+^ macrophages at the late-stage was much lower than that of CD163^+^/CD204^+^ macrophages, indicating that CD204^+^ macrophages may have functional properties towards M2 polarization. It is reported that MHC class II antigen expression may be included in the M1 macrophage pathway, whereas M2 macrophage phenotype is also associated with scavenger receptors such as CD204 [1,29].

Besides macrophage infiltration, the recruitment of lymphocytes is another important event in renal interstitial fibrosis [22]. CD4^+^ T cells were increased on days 3 to 20, with the plateau on day 9 onwards. On day 9 onwards, MHC class II^+^ macrophages began to be concomitantly increased. More interestingly, CD4^+^ T cells and MHC class II^+^ macrophages tended to appear around the damaged renal tubules at the late-stage. Because CD4^+^ T cells stimulate macrophages to generate inflammatory mediators [1], and MHC class II^+^ cells are known to present the processed antigens to CD4^+^ T cells, CD4^+^ T cells may act with the relation of MHC class II^+^ macrophages with the polarization into M1 macrophages. In fact, it is reported that CD4^+^ T cells contribute to the progress of renal fibrosis [30,31].

CD8^+^ T cells also began to be increased from day 9 onwards. M2 macrophages are reported to suppress CD8^+^ T cell-mediated cytotoxicity to tubular cells by producing anti-inflammatory cytokines [26]. The appearance of M2 macrophages may be a pathological event against the cytotoxicity by CD8^+^ T cells. Apparently, the present progressive renal fibrosis seen at the late-stage may be developed by the attendance of complicated cell types: presumably, cooperation between M1 macrophages and CD4^+^ T cells, and confrontation between M2 macrophages and CD8^+^ T cells.

The number of CD20^+^ B cells was transiently increased at the mid-stage on days 5 and 7. A study on mouse unilateral ureteral obstruction model showed that the early-stage accumulation of B cells in the affected kidney accelerated monocyte/macrophage mobilization and infiltration, leading to aggravated fibrosis [32]. The significance of CD20^+^B cells remains to be investigated because the number of CD20^+^ B cells was much lower, as compared with T cells reacting to CD4 and CD8.

In conclusion, the present study demonstrated the possible pathogenesis of CDDP-induced rat renal interstitial fibrosis, based on the immunohistochemical analyses of M1/M2 macrophage polarization and appearance of lymphocytes. CD68^+^ M1 macrophages appeared with increased M1-related factors relating to tissue damage. In progressive renal interstitial fibrosis developed at the late-stage after tissue damage, an increased number of CD163^+^ M2 macrophages were consistently seen, and there were macrophages reacting simultaneously to both CD68 and CD163; furthermore, MHC class II^+^ macrophages showed a polarization into M1 macrophages, whereas CD204^+^ macrophages were toward M2 polarization. Along with M1/M2 macrophages, CD4^+^ and CD8^+^ T cells showed an increase at the late-stage. The progressive fibrosis may be developed by complicated functions that arose via the interaction of M1/M2 macrophages (inflammatory M1 and anti-inflammatory M2) and T cells reacting to CD4 (for helper) and CD8 (for cytotoxicity). It is interesting to investigate more detailed functions of macrophages and lymphocytes, as well as the mutual relationship of these cells, in progressive renal fibrosis, particularly by using the technique of macrophage depletion, because there were somewhat differences in the distribution and kinetics of these inflammatory cells; studies on inflammatory cells would contribute to therapeutic strategies for CKD. This is the first trial of detained analyses (such as M1/M2 macrophage polarization) of inflammatory cells in vivo in CDDP-induced rat renal fibrosis.

## Figures and Tables

**Figure 1 cells-10-00257-f001:**
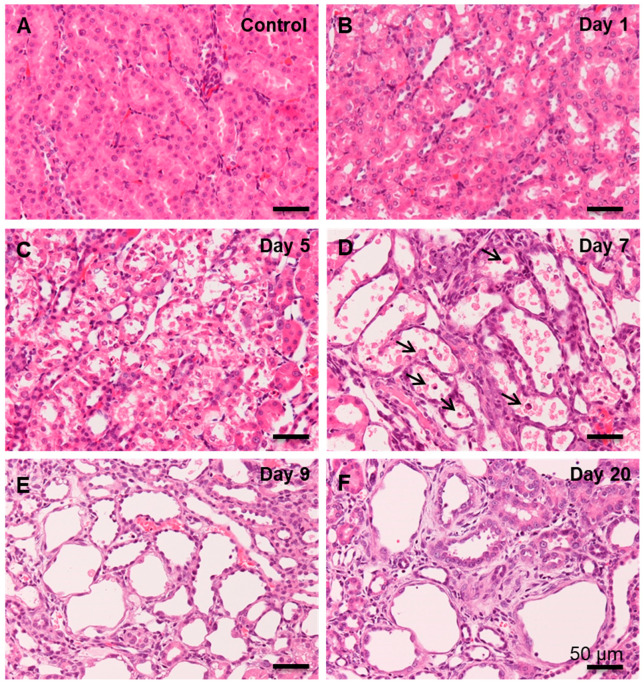
Histopathology of cisplatin (CDDP)-induced rat renal lesions. (**A**) In the control kidneys (fibrosis degree, −), no significant change is observed. (**B**) On day 1 (fibrosis degree, −) following CDDP injection, renal proximal tubular epithelial cells in the corticomedullary junction show nuclear alterations. (**C**) On day 5 (fibrosis degree, 1+), the renal proximal tubular epithelial cells show swelling and desquamation. (**D**) On day 7 (fibrosis degree, 2+), the affected tubules are dilated, showing complete desquamation, and lined partly by regenerating epithelial cells; infiltration of inflammatory cells (arrow) are seen in the injured lumina. (**E**) On day 9 (fibrosis degree, 3+), fibrosis is observed in the interstitium around the affected tubules with variously dilated lumen. (**F**) On day 20 (fibrosis degree, 4+), the damaged renal tubules lined by incompletely regenerating epithelial cells and fibrosis are seen around these tubules. HE stain, bar = 50 µm.

**Figure 2 cells-10-00257-f002:**
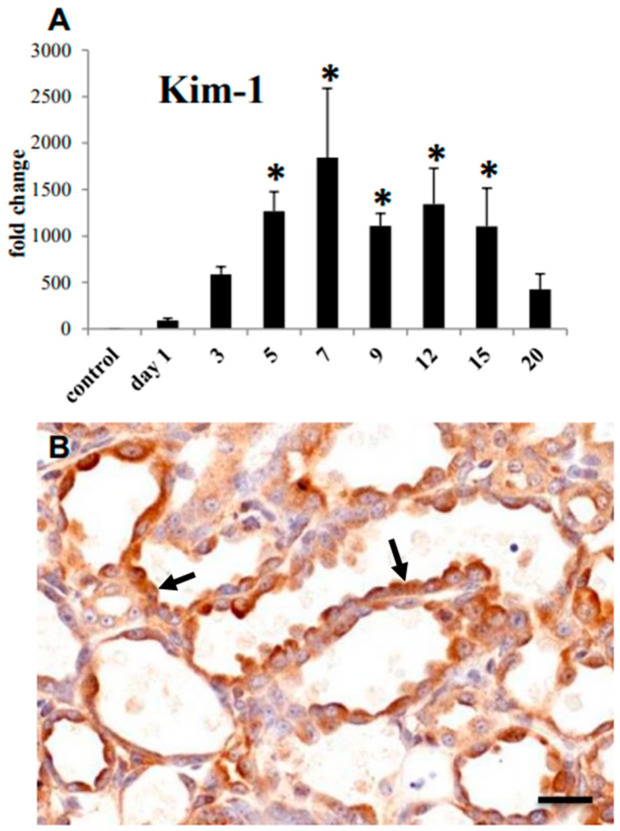
CDDP-induced renal lesions in rats. (**A**) mRNA expression of kidney injury molecule-1 (KIM-1) in the control and CDDP-injected rats on days 1 to 20; Dunnett’s test, *, significantly different from controls at *p* < 0.05. (**B**) Immunohistochemistry for KIM-1; arrows indicated immune-positive regenerating epithelial cells of the affected tubules on day 9. Bar = 50 µm.

**Figure 3 cells-10-00257-f003:**
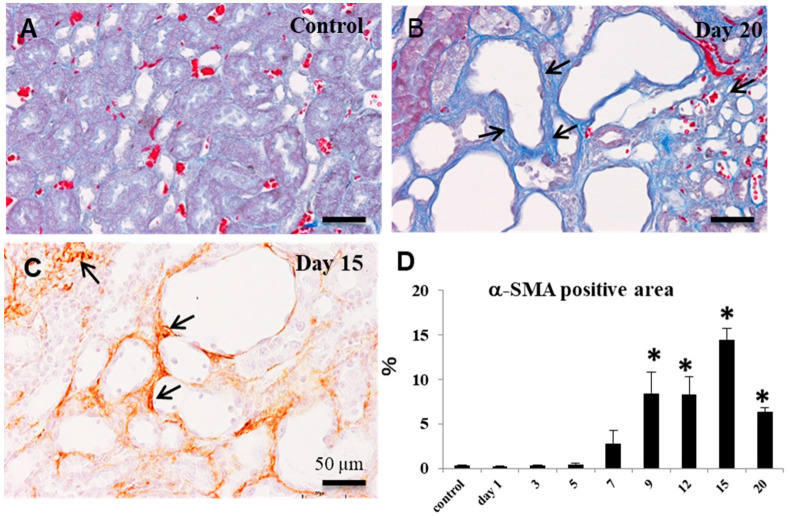
Evaluation of interstitial fibrosis of CDDP-induced rat renal lesions. (**A**) The azan–Mallory stain; in the control kidneys, collagen fibers are not seen in the interstitium (fibrosis degree, −). (**B**) The azan–Mallory stain; on day 20, severe fibrosis (fibrosis degree, 4+) (arrows) is seen in the interstitium. (**C**) Immunohistochemistry for α-smooth muscle action (α-SMA); α-SMA-positive myofibroblasts are seen in the increased fibrosis (fibrosis degree, 4+) (arrows). Bar = 50 µm. (**D**) The percentage of α-SMA-positive area in the control and CDDP-injected rats on days 1 to 20; Dunnett’s test, *, significantly different from controls at *p* < 0.05.

**Figure 4 cells-10-00257-f004:**
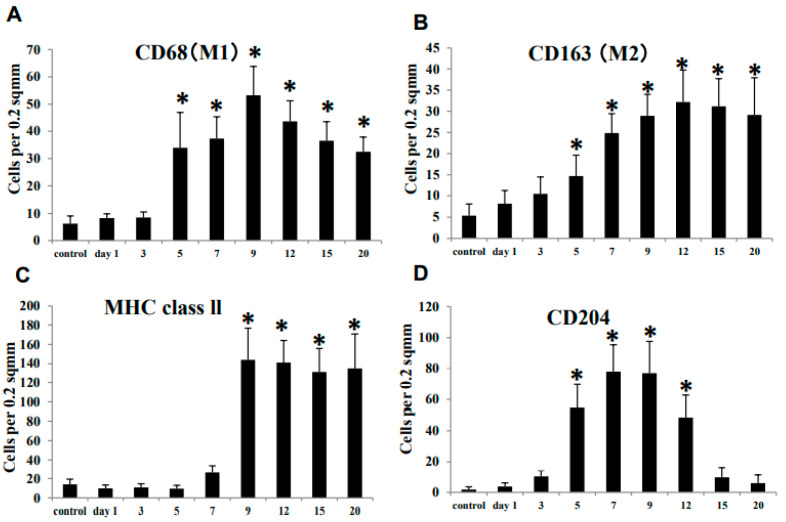
The number of macrophages reacting to CD68 (**A**, for M1), CD163 (**B**, for M2), MHC class II (**C**), and CD204 (**D**) in the control and CDDP-treated kidney on days 1–20. Note that the kinetics of these macrophages in comparison to controls differ from each other. Dunnett’s test, *, significantly different from controls at *p* < 0.05.

**Figure 5 cells-10-00257-f005:**
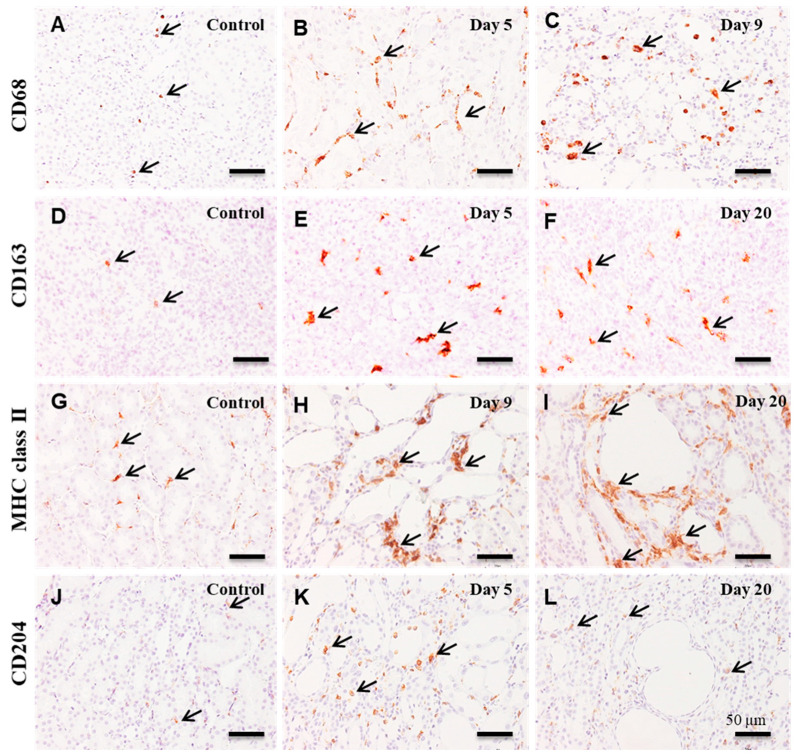
Immunohistochemistry for CD68 (**A**–**C**), CD163 (**D**–**F**), MHC class II (**G**–**I**), and CD204 (**J**–**L**) in the control and CDDP-treated kidneys. A few CD68^+^ macrophages (arrows) are seen in the corticomedullary junction of control kidneys (**A**), whereas many macrophages reacting CD68 (arrows) are present around the affected renal tubules on days 5 (**B**) and 9 (**C**). In contrast to control (**D**), many CD163^+^ macrophages (arrows) are seen on days 5 (**E**) and 20 (**F**). Increased numbers of MHC class II^+^ macrophages (arrows) are seen exclusively around the affected renal tubules on days 9 (**H**) and 20 (**I**), in contrast to control (**G**). Only a few CD204^+^ macrophages are present in the control (**J**) and on day 20 (**L**), whereas, on day 5, CD204^+^ macrophages are increased (**K**). Note that there are differences in the distribution and location of these macrophages from each other. Counterstained with hematoxylin. Bar = 50 µm.

**Figure 6 cells-10-00257-f006:**
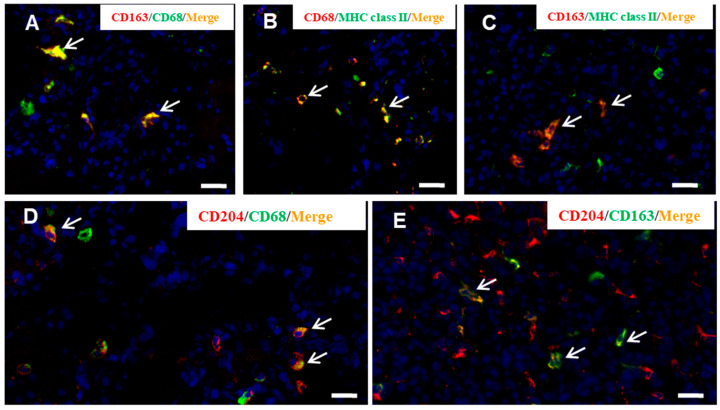
Double immunofluorescence for CD163/CD68 (**A**), CD68/MHC class II (**B**), CD168/MHC class II (**C**), CD68/CD204 (**D**), and CD163/CD204 (**E**) in CDDP-induced renal lesions on day 9. Yellow color (arrows) indicates double-positive reactions. Blue indicates nuclei stained with 4′,6-diamino-2-phenylindole (DAPI). Bar = 20 µm.

**Figure 7 cells-10-00257-f007:**
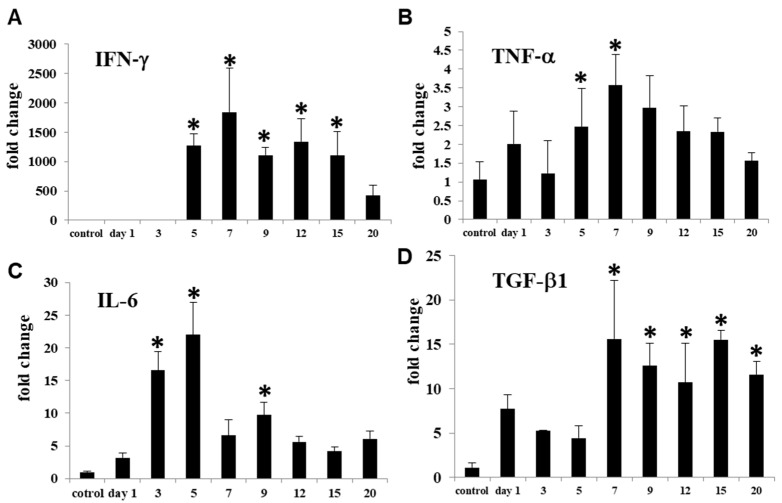
mRNA expressions for M1-related factors such as IFN-γ (**A**), TNF-α (**B**) and IL-6 (**C**), and for M2-related factor TGF-β1 (**D**) in the control and CDDP-treated kidneys on days 1–20. Expression levels were normalized byβ-actin RNA level. Dunnett’s test, *, significantly different from controls at *p* < 0.05.

**Figure 8 cells-10-00257-f008:**
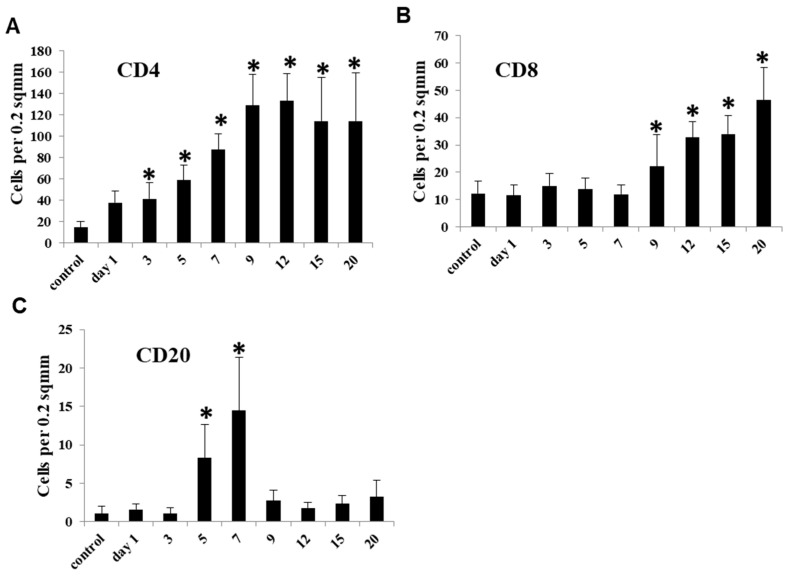
The number of lymphocytes reacting to CD4 (**A**), CD8 (**B**) and CD20 (**C**) in the control and CDDP-treated kidneys on days 1–20. Note that the kinetics of these lymphocytes to controls differs from each other. Dunnett’s test, *, significantly different from controls at *p* < 0.05.

**Figure 9 cells-10-00257-f009:**
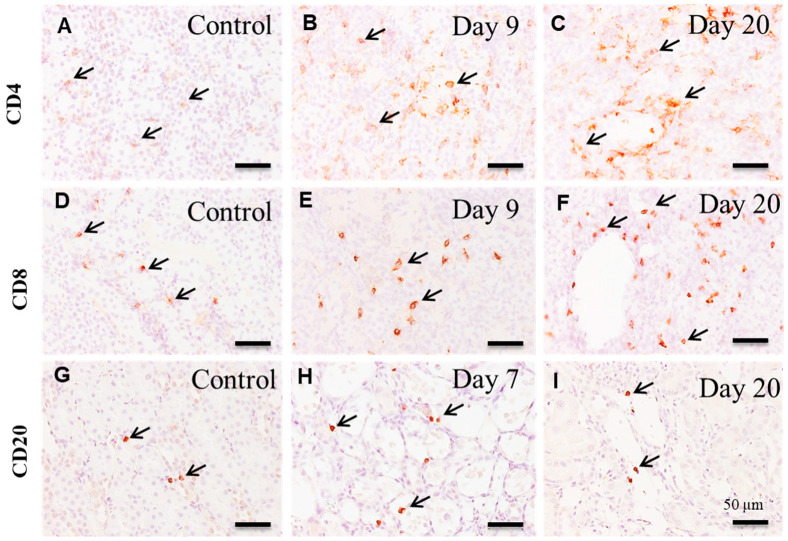
Immunohistochemistry for CD4 (**A**–**C**), CD8 (D-F) and CD20 (**G**–**I**) in the control and CDDP-treated kidneys. In contrast to control (**A**), the number of CD4^+^ T lymphocytes(arrows) are increased on day 9 (**B**) and aggregated CD4^+^ T cells (arrows) are seen around the affected renal tubules on day 20 (**C**). In contrast to control (**D**), CD8^+^ T cells (arrows) are increased on day 9 (**E**) and tend to aggregate around the affected renal tubules on day 20 (**F**). In contrast to control (**G**) and on day 20 (**I**), CD20^+^ B cells (arrows) are increased on day 7 (**H**). Note that the distributions of these lymphocytes are different from each other. Counterstained with hematoxylin. Bar = 50 µm.

**Table 1 cells-10-00257-t001:** Primary antibodies used for immunohistochemistry and immunofluorescence.

Antibody	Type	Dilution	Pretreatment	Source
CD163 (ED2)	Mouse monoclonal	1/300	-	AbDSerotec, Oxford, UK
CD68 (ED1)	Mouse monoclonal	1/500	10 µg/mL proteinase K, 20 min	Merck Millipore, MA, USA
MHC class II (OX-6)	Mouse monoclonal	1/300	Microwaving in citrate buffer, 20 min	AbDSerotec, Oxford, UK
CD204 (SRA-E5)	Mouse monoclonal	1/500	Microwaving in citrate buffer, 20 min	Transgenic Inc., Kumamoto, Japan
α-SMA	Mouse monoclonal	1/1000	-	Dako, CA, USA
CD4	Mouse monoclonal	1/2000	-	AbDSerotec, Oxford, UK
CD8	Mouse monoclonal	1/200	-	AbDSerotec, Oxford, UK
CD20	Goat polyclonal	1/200	Microwaving in citrate buffer, 20 min	Santa Cruz Biotechnology Inc., TX, USA
KIM-1	Goat polyclonal	1/500	-	Immunology Consultants Laboratory, OR, USA

**Table 2 cells-10-00257-t002:** Real-time RT–PCR primers.

Primer	NCBI Symbol	Forward	Reverse
IFN-γ	*Ifng*	tcgcacctgatcactaacttcttc	cgactccttttccgcttcc
TNF-α	*Tnf*	tgcctcagcctcttctcattc	gctcctctgcttggtggttt
TGF-β1	*Tgfb1*	cttcagctccacagagaagaactgc	cacgatcatgttggacaactgctcc
KIM-1	*Havcr1*	gtctgtattgttgccgagtgga	gttgtgggtcttgttggagga
β-actin	*Actb*	taaagacctctatgccaacac	ctcctgcttgctgatccacat

IFN-γ—interferon-γ, TNF-α—tumor necrosis factor-α, TGF-β1—transforming growth factor-β1, KIM-1—kidney injury molecule-1, NCBI—The National Center for Biotechnology Information.

## Data Availability

All data are contained within the article.

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
