# Peer review of "Immunophenotypical Characterization of M1/M2 Macrophages and Lymphocytes in Cisplatin-Induced Rat Progressive Renal Fibrosis"

_cells, 2021, doi:10.3390/cells10020257_

Round 1

Reviewer 1 Report

Comments to the Authors

to get a snapshot of the functional profile of M1/M2 macrophages during CKD, the authors should perform macrophage ( M1/M2) purification from injured kidneys followed by RNAseq.

As this would be too much for all time points, I suggest to perform this experiment for time point 5-days (high influx M1/M2 but prior fibrosis) and at day 12 (active fibrogenesis), isolate CD45 cells with MACS and then sort for CD68 MHCII or CD163 CD204+. kidneys can be pooled to obtained enough cells for RNA seq.

Reviewer 2 Report

This study  analyses some information about renal fibrosis in relation to the role of macrophages and myofibroblasts, an important study as renal fibrosis can lead to chronic kidney disease.

This research does not reach certain conclusions, but it may contribute to further studies to arrive at suitable therapeutic strategies for CKD in the future.

In particular:

Pay attention to formatting, spaces for example. There are numerous errors.

Why did the authors prefer to perform the Duncan test rather than ANOVA followed by an appropriate test? Now the difference between the days does not appear.

Reviewer 3 Report

Nakagawa et al. aimed to investigate the roles of M1/M2 macropahges and lymphocyte in the progression of cisplatin-induced renal fibrosis model. They found that interstitial fibrosis and increased infiltration of CD163+ M2 macrophages and T lymphocytes developed in the late stage of cisplatin nephropathy model. Although the results might be interesting, I have several concerns about the study.

1. The contributing role of M1 and M2 macrophages in the development of interstitial fibrosis in this study is not clear. It will be more informative to analyze the extent of fibrosis after depletion of specific macrophage subsets.
2. The interaction of macrophages and lymphocytes is not explored.
3. The number of animals per group is only three and may be inadequate for detection of significant biological difference.
4. There is no quantification of many histological and immunohistochemical experiments.

Round 2

Reviewer 1 Report

Dear Authors, I understand the issues that you would encounter in repeating the whole animal experiments (both for the time-consuming part and the concerns from animal ethic committee).

I would like to ask you to perform instead a triple staining at time-points day 5 and day 9 or day12 for CD68, CD163 and TGFb1 , and a triple IHC-P for CD68, CD163 and IL6 (or TNFa). It would be interesting to verify in your experimental settings the functionality of this M1/M2 "hybrid" macrophages, besides showing the M2-driven profibrotic environment in injured kidneys.

Reviewer 3 Report

1. Given that alpha-SMA is also expressed in the renal arterioles, quantification and statistical analyses of the H&E-stained and the azan Mallory-stained histological photomicrographs are required.

2. In the line 16, page 6 of the section 3.2, the authors claimed a peak of alpha-SMA-positive area occurred on day 16. This statement is not supported by the statistical analysis because Dunnett's test compares each of the treatment groups against one single control group. The Dunnett's test does not compare the groups other than the control group among themselves at all (PMID: 22420233; PMID: 30157585). Please clarify this point by using other multiple comparison test, such as Tukey's test. Similar condition is also noted on the line 14 of the page 7.

3. The nomenclature of gene in the PCR experiments is incorrect.

4. No arrow is found in the figure 2B.

5. In Figure 3, IHC of alpha-SMA for control group should also be shown.

6. The sizes of panels in the Figure 6 are unequal. Scale bars should be provided in the each panel.

7. Figure 8 seems to be the quantification of Figure 9. To increase the readability, please combine the Figure 8 and Figure 9 into a single figure.

8. There are several typographical errors and the grammar can be improved.
